# The National Early Warning Score 2 with Age and Body Mass Index (NEWS2 Plus) to Determine Patients with Severe COVID-19 Pneumonia

**DOI:** 10.3390/jcm13010298

**Published:** 2024-01-04

**Authors:** Konlawij Trongtrakul, Pattraporn Tajarernmuang, Atikun Limsukon, Theerakorn Theerakittikul, Nutchanok Niyatiwatchanchai, Karjbundid Surasit, Pimpimok Glunriangsang, Chalerm Liwsrisakun, Chaiwat Bumroongkit, Chaicharn Pothirat, Juthamas Inchai, Warawut Chaiwong, Panida Chanayat, Athavudh Deesomchok

**Affiliations:** 1Division of Pulmonary, Critical Care, and Allergy, Department of Internal Medicine, Faculty of Medicine, Chiang Mai University, Chiang Mai 50200, Thailand; konlawij.tr@cmu.ac.th (K.T.); pattraporn.t@cmu.ac.th (P.T.); atikun.limsukon@cmu.ac.th (A.L.); theerakorn.t@cmu.ac.th (T.T.); nutchanok.n@cmu.ac.th (N.N.); chalerm.liw@cmu.ac.th (C.L.); chaiwat.b@cmu.ac.th (C.B.); chaicharn.p@cmu.ac.th (C.P.); juthamas.i@cmu.ac.th (J.I.); warawut.chai@cmu.ac.th (W.C.); panida.kanjanavaha@cmu.ac.th (P.C.); 2Nakornping Hospital, Chiang Mai 50180, Thailand; offkarj1@hotmail.com; 3Sansai Hospital, Chiang Mai 50290, Thailand; pearl090@yahoo.com

**Keywords:** severe COVID-19 pneumonia, critical COVID-19 illness, early determination, severity prediction score, National Early Warning Score 2 (NEWS2), NEWS2 Plus

## Abstract

(1) Background: Early identification of severe coronavirus disease 2019 (COVID-19) pneumonia at the initial phase of hospitalization is very crucial. To address this, we validated and updated the National Early Warning Score 2 (NEWS2) for this purpose. (2) Methods: We conducted a study on adult patients with COVID-19 infection in Chiang Mai, Thailand, between May 2021 and October 2021. (3) Results: From a total of 725 COVID-19 adult patients, 350 (48.3%) patients suffered severe COVID-19 pneumonia. In determining severe COVID-19 pneumonia, NEWS2 and NEWS2 + Age + BMI (NEWS2 Plus) showed the C-statistic values of 0.798 (95% CI, 0.767–0.830) and 0.821 (95% CI, 0.791–0.850), respectively. The C-statistic values of NEWS2 Plus were significantly improved compared to those of NEWS2 alone (*p* = 0.012). Utilizing a cut-off point of five, NEWS2 Plus exhibited better sensitivity and negative predictive value than the traditional NEWS2, with values of 99.7% vs. 83.7% and 98.9% vs. 80.7%, respectively. (4) Conclusions: The incorporation of age and BMI into the traditional NEWS2 score enhanced the efficacy of determining severe COVID-19 pneumonia. Physicians can rely on NEWS2 Plus (NEWS2 + Age + BMI) as a more effective decision-making tool for triaging COVID-19 patients during early hospitalization.

## 1. Introduction

Coronavirus disease 2019 (COVID-19), caused by severe acute respiratory syndrome coronavirus 2 (SARS-CoV-2), first emerged in December 2019 [1]. Since then, COVID-19 has spread around the world and is considered a global pandemic [2]. COVID-19 is attributed to several organ dysfunctions such as those affecting the respiratory tract [3,4], cardiovascular system [5,6], gastrointestinal tract [7], and cutaneous manifestation [8] and coagulopathy [9]. Indeed, focusing on respiratory tract involvement, a mortality rate was found to be greater in patients with severe COVID-19 pneumonia than those with non-severe, approximately 20% vs. 0.6%, respectively [3]. Additionally, a higher mortality of greater than 50% was found in patients suffering from COVID-19-associated acute respiratory distress syndrome [4].

A rapidly escalating number of patients with COVID-19 required an overwhelming number of healthcare resources in a short period, obligating healthcare workers to develop a prompt triaging tool for early identification of those patients with severe respiratory conditions. Some studies supported the utilization of the National Early Warning Score 2 (NEWS2) to determine severe COVID-19 [10,11,12]. Nonetheless, other significant risk factors were involved in the severe COVID-19 conditions as well, such as aging [3,13,14,15,16], male gender [3,16], obesity [17,18], diabetes mellitus [3,13,17], and hypertension [13].

Therefore, we aimed to validate and update NEWS2 in conjunction with other significant prognostic factors at hospitalization in determining those patients with severe COVID-19 pneumonia. 

## 2. Materials and Methods

### 2.1. Study Design

We retrospectively analyzed the information from prospective data collected from the registry for COVID-19 patients from three hospitals in Chiang Mai, Thailand, which were Sansai Hospital (a secondary-care hospital), Nakornping Hospital (a tertiary-care hospital), and Maharaj Nakorn Chiang Mai Hospital (a tertiary-care and a university-base hospital). 

All adult patients who were hospitalized with SARS-CoV-2 pneumonia from May 2021 to October 2021 were involved. The registry was approved by the Research Ethics Committee of the Faculty of Medicine, Chiang Mai University, Thailand (study code: MED-2564-08109, approved on 3 May 2021). In addition, the study was registered in the Thai Clinical Trials Registry (study ID: TCTR20210827005, approved on 27 August 2021). The study was performed in accordance with the Declaration of Helsinki, a statement of ethical principles for medical research involving human subjects. The Research Ethics Committee of the Faculty of Medicine, Chiang Mai University, Thailand, waived the requirement for written informed consent as the research involved no more than minimal risk and was performed as a secondary data analysis. In addition, the information used in the study was anonymized. 

### 2.2. Inclusion and Exclusion

Patients aged 18 or older and hospitalized with a confirmed positive reverse transcription-polymerase chain reaction (RT-PCR) result for SARS-CoV-2 were identified from nasal or pharyngeal swab specimens [19,20]. Eligible participants required a diagnosis of COVID-19 pneumonia based on the WHO interim guidance [21], where a positive pulmonary infiltration was present on chest imaging. We restricted data recording to when patients only presented with the most severe COVID-19 pneumonia, even if they were transferred across centers. We excluded COVID-19 patients who presented with a localized upper respiratory tract infection without any evidence of pneumonia involved in the study.

### 2.3. Definitions for Severe COVID-19 Pneumonia

A patient diagnosed with severe COVID-19 pneumonia was defined according to the WHO Working Group on the Clinical Characterization and Management of COVID-19 infection ordinal scale from six to nine [22]. This included patients who were hospitalized and treated with 1) a high-flow nasal cannula (HFNC) or non-invasive ventilation (NIV), 2) intubation and mechanical ventilation (IMV) with PaO2/FiO2 ratio (PF ratio) ≥ 150 or SpO2/FiO2 ≥ 200, 3) IMV with PF ratio < 150 (SpO2/FiO2 < 200) or vasopressor therapy, and 4) IMV with PF ratio < 150 and vasopressor therapy, dialysis, or extracorporeal membrane oxygenation (ECMO) [22].

### 2.4. Clinical and Laboratory Investigations

The patients’ baseline demographics obtained in the study included the patient demographics: age, sex, body mass index (BMI); current smoking status; pre-existing comorbidities: hypertension, diabetes mellitus, dyslipidemia, respiratory diseases (asthma, chronic obstructive pulmonary disease, among others), chronic kidney disease, heart diseases (coronary artery disease, chronic heart failure, among others), and others (liver cirrhosis, malignancy, and immunocompromised disease); onset of illness before hospitalization; vital signs on admission: body temperature, heart rate, respiratory rate, systolic blood pressure, diastolic blood pressure, and pulse oximetry (SpO2) at room air; and disease severity at hospital admission assessed by NEWS2. 

In addition, the initial laboratory investigations at hospital admission were also conducted, which consisted of analysis of hemoglobin, white blood cell count, absolute lymphocyte count, platelet, blood urea nitrogen, creatinine, and inflammatory markers including D-dimer, c-reactive protein, procalcitonin, lactate, lactate dehydrogenase, and erythrocyte sedimentation rate. Multi-lobar pneumonia was defined as a positive pulmonary infiltration of more than one lobe on chest radiographic abnormality.

Antiviral treatment (Favipiravir and Remdesivir) and steroids (systemic and oral route) were prescribed according to Thai Guidelines on Clinical Practice, Diagnosis, Treatment and Prevention of Healthcare-Associated Infection for COVID-19 [23]. In addition, vasopressor prescription and salvage therapy with an interleukin-6 receptor antagonist, named Tocilizumab, were also collected. 

### 2.5. Study Outcomes

The primary outcome was progression to a severe form of COVID-19 pneumonia, defined according to the WHO Working Group on the Clinical Characterization and Management of COVID-19 infection, especially in patients using respiratory support [22]. Thereafter, the impact of severe COVID-19 pneumonia was compared with the non-severe group. Other outcomes, including ICU mortality, hospital mortality, ICU length of stay, and hospital length of stay, were obtained, and other complications during hospitalization, including acute respiratory distress syndrome (ARDS) requiring IMV, acute kidney injury (AKI), hospital-acquired pneumonia (HAP)/ventilator-associated pneumonia (VAP), and pulmonary embolism, were also gathered.

### 2.6. Statistical Analysis

Continuous variables were expressed as mean and standard deviation (SD) or median and interquartile ranges (IQRs), as appropriate. Categorical variables were expressed as counts and percentages. A comparison of continuous variables between the patients with severe COVID-19 pneumonia and those without were analyzed using Student’s t-test or the Wilcoxon rank sum test, as appropriate, while a comparison of categorical variables was analyzed using Fisher’s exact test.

#### 2.6.1. Determination of Candidate Prognostic Factors 

We performed univariable logistic regression (LR) analysis to estimate the effect size of the association of NEWS2 with severe COVID-19 pneumonia, reporting with the odds ratio (OR) and 95% confidence interval (95% CI). Additionally, we employed LR to examine patient characteristics that were potential indicators of severe COVID-19 pneumonia without relying on laboratory investigations. A final multivariable LR analysis was performed, with a selection criterion of significant variables in univariable LR with a *p*-value of less than 0.05.

#### 2.6.2. NEWS2 and NEWS2 Plus Scoring Assignment

Each independent variable’s score was calculated by dividing its coefficient by the lowest coefficient in the multivariable LR model [24]. The score was rounded to the nearest integer. The NEWS2 Plus scoring system was integrated with scores from NEWS2 and significant parameters. Combining traditional NEWS2 with assigned scores for significant variables generated the NEWS2 Plus score. 

#### 2.6.3. Model Performances of NEWS2 and NEWS2 Plus

The model discrimination between cases with severe and non-severe COVID-19 pneumonia was assessed using the C-statistic [25]. In addition, the C-statistic of each NEWS2 Plus model was compared with the traditional NEWS2. We also analyzed integrated discrimination improvement (IDI) [26] and net reclassification improvement (NRI) [27,28] to quantify the model performance. 

Model calibration was evaluated, using a calibration intercept, calibration plot, calibration in the large (CITL), and calibration slope (CS). In addition, we internally validated the model using a bootstrapping procedure with 1000 replications.

Diagnostic performance for each model, including sensitivity, specificity, positive predictive value (PPV), and negative predictive value (NPV), was reported accordingly. 

All *p*-values were two-tailed, and a *p*-value < 0.05 was considered to be statistically significant. All statistical analyses were performed using STATA version 16.0 (Stata Corp LP, College Station, TX, USA).

## 3. Results

### 3.1. Demographic and Clinical Features

A total of 725 cases (Sansai Hospital, *n* = 64; Nakornping Hospital, *n* = 315; and Maharaj Nakorn Chiang Mai Hospital, *n* = 346) were involved in the study. Of these, 350 (48.3%) cases involved severe COVID-19 pneumonia. Table 1 demonstrates the patients’ demographics and clinical features between groups. There were older, predominantly male patients with greater BMI and more hypertension, diabetes mellitus, and chronic kidney disease diagnoses in the severe group than in the non-severe group. A slightly longer median onset of illness before hospitalization in the severe group than in the non-severe group was found (4 [IQR 2, 6] days vs. 3 [IQR 2, 5] days, respectively, *p* = 0.005), with greater severity of illness measured by NEWS2 (6 [IQR 4, 9] vs. 3 [IQR 2, 5], respectively, *p* < 0.001). Additionally, the severe group had less leukocytosis and less ALC but greater levels of inflammatory markers, including D-dimer, CRP, procalcitonin, LDH, and ESR. Furthermore, there were more patients with multi-lobar involvement in chest imaging in the severe group (Table 1).

### 3.2. Treatment, Complications, and Outcomes of the Patients

Most severe COVID-19 patients received Remdesivir, systemic corticosteroids, rescue therapy with Tocilizumab, and more oxygen support via an HFNC and IMV (Table 2). The severe group had more organ dysfunction and complications during hospitalization in terms of ARDS, AKI, HAP/VAP, and pulmonary embolism than the non-severe group (Table 2). There were also greater ICU mortality and hospital mortality and longer ICU length of stay and hospital length of stay in the severe group than in the non-severe group significantly (Table 2).

### 3.3. Prognostic Factors for Severe COVID-19 Pneumonia: NEWS2 and Others

The results indicated an almost 1.5-fold increase in the risk for severe COVID-19 pneumonia when NEWS2 increased per one point, with the univariable OR and multivariable OR of 1.523 (95% CI, 1.424–1.629, *p* < 0.001) and 1.459 (95% CI, 1.361–1.565, *p* < 0.001), respectively (Table 3). Other significant baseline prognostic factors to determine the severe group were age and BMI (Table 3). Age was associated with the severe group with the univariable OR and multivariable OR of 1.044 (95% CI, 1.034–1.054, *p* < 0.001) and 1.040 (95% CI, 1.028–1.052, *p* < 0.001), respectively (Table 3). In addition, BMI was associated with the severe group with the univariable OR and multivariable OR of 1.042 (95% CI, 1.018–1.067, *p* = 0.001) and 1.077 (95% CI, 1.044–1.110, *p* < 0.001), respectively (Table 3). Therefore, we added these two prognostic factors for developing the NEWS2 Plus models.

### 3.4. NEWS2 and NEWS2 Plus Models to Determine Severe COVID-19 Pneumonia

We treated NEWS2 as its traditionally calculated value and categorized age and BMI into three levels and assigned the score for each level in order to make it more convenient to utilize at the bedside. The assigned scores for the patient’s age, below 40, 40 to 59, or 60 or greater, were 0, 3, and 4 points, respectively (Table 4). The assigned scores for normal weight (BMI of 24.9 kg/m^2^ or lower), overweight (BMI of 25.0 to 29.9 kg/m^2^), and obesity (BMI of 30.0 kg/m^2^ or greater) were 0, 2, and 3 points, respectively (Table 4). Accordingly, the traditional maximum score for NEWS2 is 20. After adding age and BMI, the maximal total score for NEWS2 + Age + BMI was 27 (20 + 4 + 3).

### 3.5. The Performance of NEWS2 and NEWS2 Plus Models

Table 5 demonstrates the discriminative ability of the traditional NEWS2 with other models. We found that in determining the severe group, the traditional NEWS2 had a C-statistic value of 0.798 (95% CI, 0.767–0.830). NEWS2 + Age and NEWS2 + BMI models exhibited a better discriminative ability than the traditional NEWS2. However, NEWS2 + Age + BMI had the best discrimination ability, with a C-statistic value of 0.821 (95% CI, 0.791–0.850). When comparing the NEWS2 + Age + BMI model with the traditional NEWS2 model, we found a significant difference in C-statistic value at 0.022 (95% CI, 0.005–0.040). 

Furthermore, the NEWS2 + Age + BMI model provided the most favorable improvement in discrimination and reclassification when compared with the traditional NEWS2 model, with an IDI and NRI of 5.3% (95% CI, 2.9–8.7%) and 45.7% (95% CI, 32.1–65.0%), respectively (Table 5). 

The calibration intercept, CITL, CS, and internally validated values are reported in Table 5 and Figure 1. The internally validated C-statistic values utilizing the bootstrapping method with 1000 replications for traditional NEWS2 and NEWS2 + Age + BMI were 0.797 (95% CI, 0.766–0.831) and 0.821 (95% CI, 0.792–0.852), respectively (Table 5). Additionally, the observed outcome was systemically determined by the predicted risks (probability) in all apparent models when classified from the individualized risk score (Figure 2).

### 3.6. Diagnostic Performance of NEWS2 and NEWS2 Plus for Severe COVID-19 Pneumonia

From the previous study, the traditional NEWS2 cut-off point in determining patients with severe COVID-19 pneumonia was five points [29]. We applied this cut-off to all models and calculated the corresponding diagnostic performance. The traditional NEWS2 provided a sensitivity and NPV of 83.7% and 80.7%, respectively (Table 6). The cut-off point of five for NEWS2 + Age + BMI delivered the highest sensitivity and NPV than the traditional NEWS2, with a value of 99.7% and 98.9%, respectively (Table 6).

## 4. Discussion

The overwhelming number of emerging cases of severe COVID-19 creates a lack of resources for all patients, even in countries with well-organized healthcare systems [30,31,32]. Early identification of patients with severe COVID-19 pneumonia is crucial for providing patients with proper and effective treatment. Therefore, the aim of this study was to validate NEWS2 in our setting and integrate other significant prognostic factors at the time of hospitalization into the NEWS2 score to enhance its ability to assess severe COVID-19 pneumonia. 

NEWS2 was chosen based on the fact that it contains only physiological parameters, including respiratory rate, SpO2 scale for acute hypoxemic and acute hypercapnic respiratory failure, oxygen supplementation, systolic blood pressure, heart rate, consciousness, and body temperature, without any laboratory or imaging results [33]. In terms of severe COVID-19, we defined this condition according to the WHO original scale from six to nine [22].

Through regression analyses, we identified associations between age and BMI with severe COVID-19. Consequently, we developed the NEWS2 Plus model and compared its performance with the traditional NEWS2 score. 

The traditional NEWS2 and NEWS2 Plus (NEWS2 + Age + BMI) disclosed those patients with severe COVID-19 pneumonia to a reasonable degree, with a C-statistic of 0.798 (95% CI, 0.767–0.830) and 0.821 (95% CI, 0.791–0.850), respectively. There was a significantly better C-statistic (*p* = 0.022) from the new model when compared to the traditional NEWS model, with enhanced IDI (5.3%; 95% CI, 2.9–8.7%) and improved NRI (45.7%; 95% CI, 32.1–65.0%). All appearance model results and internal validation calibrations (calibration intercept, CITL, and CS) were acceptable. At an acceptable cut-off of five points [29], the diagnostic performance in terms of sensitivity and NPV was the highest among the NEWS2 + Age + BMI model over other models. 

Some evidence supported that advanced age and obesity were strongly associated with more severe COVID-19 [3,13,14,15,17,18]. Therefore, a modified version of NEWS named NEWS-C has been previously proposed by Liao et al., where an additional three points are assigned to the NEWS score for patients with an age greater than 65 [34].

In one study, the application of NEWS-C yielded better performance than using NEWS alone [35]. However, it was contradicted in another study, which revealed no advantage of NEWS-C over NEWS at all (AUC of 0.72 vs. 0.74, respectively) [36].

Other studies revealed the better performance of NEWS-C when compared with NEWS2 [10,37]. Su et al. revealed better accuracy of NEWS-C than that of NEWS2 in terms of detecting early deterioration of respiratory function, with the AUC of 0.79 vs. 0.59, respectively [10]. It was also better for determining the need for intensive respiratory support, with the AUC of 0.89 vs. 0.69, respectively [10]. A better use in determining the need for ICU admission was also reported in another study, with the AUC of 0.88 and 0.80, respectively [37]. 

NEWS itself does not have any SpO2 scale for patients with acute hypercapnic respiratory failure. On the other hand, NEWS2 describes the SpO2 scale separately between those patients with acute hypoxemic and acute hypercapnic respiratory failure [33]. Although the use of NEWS2 was satisfactory in determining inpatients with COVID-19 deterioration, with the AUC ranges from 0.59 to 0.90 [10,11,37,38,39], the enhanced performance of NEWS2 in combination with age and BMI remains an issue of great interest. Additionally, to the best of our knowledge, there is a lack of evidence regarding the use of NEWS2 with age and BMI, hence the motivation for this study.

Calculating NEWS2 is not only simpler but also exhibits significantly better performance than that of the more complex scoring system named COVID-GRAM (AUC of 0.87 vs. 0.77, respectively) [12]. The COVID-GRAM score contains 10 characteristics at the time of hospitalization, namely age, hemoptysis, dyspnea, unconsciousness, number of comorbidities, cancer, chest radiographic abnormality, neutrophil-to-lymphocyte ratio, LDH, and direct bilirubin [40]. However, NEWS2 provided less accuracy when compared to the COVID-19 Severity Index (AUC of 0.80 vs. 0.94, respectively) [37]. The COVID-19 Severity Index is composed of the following factors: age, sex, diabetes mellitus, dyspnea, bilateral infiltration on chest X-ray, D-dimer, lymphocyte count, and platelet in addition to NEWS2 [37]. Not only do both COVID-GRAM and COVID-19 Severity Indices contain age as their prognostic factor, but both also require laboratory and imaging results for score calculation.

Another simple tool for triggering COVID-19 deterioration is the ROX index ([SpO2/FiO2/RR]). Generally, the ROX index has been proven to determine those patients with COVID-19 who will fail HFNC therapy [41,42]. For triaging deterioration, Prower et al. demonstrated that the use of the ROX index outperformed the use of NEWS2 (AUC of 0.848 vs. 0.815, respectively) [43]. The use of instantaneous FiO2 in the ROX index may be inaccurate versus using the categorization of O2 therapy as yes/no as in the NEWS2 score. However, further investigation to prove this concept is needed. 

As highlighted earlier, aging has been demonstrated as a predictor to determine severe COVID-19. This prompts us to consider another factor that also influences severe COVID-19. Obesity stands out as a potential risk factor for severe COVID-19, primarily due to the characteristic vulnerabilities of this population. These individuals typically exhibit low cardiac and respiratory reserves and metabolic derangement and may be further endangered with immune dysregulation [44]. Some evidence supports the notion that obesity influences the severity of COVID-19. A study conducted in a French ICU demonstrated a high prevalence of obesity (BMI > 30), reaching nearly 50% among patients with COVID-19 admitted to their ICU. These populations with obesity faced a higher risk of experiencing more severe COVID-19 outcomes, with a gradually increasing likelihood of requiring IMV ranging from 10 to 15% according to the degree of obesity, categorized as normal weight (BMI < 25) to overweight (BMI 25–30), obesity (BMI 30–35), and morbid obesity (BMI > 35). Moreover, the prevalence of obesity in COVID-19 cases was markedly higher than that in non-COVID-19 cases, where the obesity prevalence was only 25%. A systematic review also revealed that obesity increased the risk of COVID-19 hospitalization and death, with ORs of 0.172 and 1.25, respectively [45]. Therefore, obesity should be considered as an important predictor for severe COVID-19 and integrated into risk assessment strategy.

Our study provided reasonable justification for using NEWS2 and NEWS2 Plus at hospitalization for determining severe COVID-19. These scores are suitable and can assist the forefront clinician decision-making process during the early phase of COVID-19 infection, helping to determine whether the patients require hospitalization or not. Importantly, the score relies solely on physiological parameters, without any need for laboratory or imaging results. Notably, adding age and BMI improved discriminative ability compared to the traditional NEWS2, since these two demographics are involved in the risk of severe COVID-19. A logistic function was utilized to identify and subsequently confirm the association. The assigned score for age and BMI was calculated according to the standard of practice. Additionally, all models were calibrated and internally validated using bootstrapping methods. For application, using the cut-off at five points, the NEWS2 + Age + BMI provided the most sensitivity and NPV (99.7% and 98.9%, respectively) than that of NEWS2 (83.7% and 80.7%). This implies that when the patient has a NEWS2 + Age + BMI score of five or greater, it enhances the discovery of cases with severe conditions. If patients have a NEWS2 + Age + BMI of less than five at the initial point of hospitalization, they are less likely to experience severe COVID-19.

## 5. Limitation

Our study has some limitations. First, because COVID-19 is an emerging disease with a varying degree of case characteristics among populations or countries, with the pace of COVID-19 waves from alpha to gramma and the changes in clinical practice over time, our NEWS2 Plus models may not guarantee accuracy for these case-mixed populations. Adaptation of a prediction model in a local setting is advised [46]. Second, a NEWS2 + Age + BMI score was developed from a registry representing only one city in Thailand. Therefore, further investigations to validate the score in terms of its generalizability are needed. Third, the categorization of age and BMI might reduce the power of continuous distribution when applied to the model. Currently, machine learning or advanced prediction models can integrate continuous parameters. However, for more convenience, categorization leads to simpler calculation at the bedside. Fourth, no sample size was calculated prior to the study. We believe that the number of cases in our registry (*n* = 725) is large enough to demonstrate the significance of the NEWS2 Plus models. Lastly, it is noteworthy that the NEWS2 score has recently been validated for determining the progression of community-acquired pneumonia [47]. However, the extension of the NEWS2 Plus beyond severe COVID-19 cases warrants future investigation to establish its applicability in diverse populations experiencing respiratory infections. This imperative arises from the fact that the NEWS2 Plus model is composed solely of physiological variables, age, and BMI. 

## 6. Conclusions

The likelihood of severe COVID-19 pneumonia could be better determined using NEWS2 + Age + BMI than with traditional NEWS2. Supported by using this score, physicians may triage from the early phase of hospitalization and even transfer patients to a better level of care or a more advanced center.

## Figures and Tables

**Figure 1 jcm-13-00298-f001:**
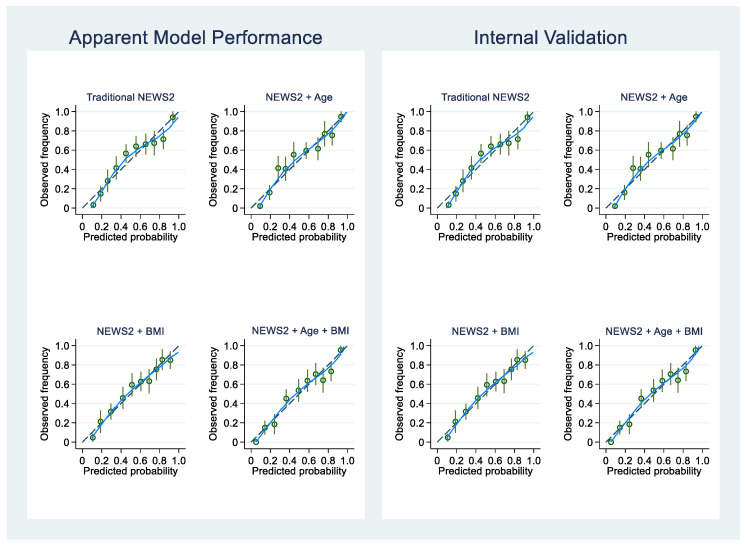
Apparent performance and internal validation of NEWS2, NEWS2 + Age, NEWS2 + Age + BMI models. The graph shows the frequency of observed outcome along with its 95% CI (green circles and vertical lines), Lowess line (continuous blue line), and a reference (black dashes line).

**Figure 2 jcm-13-00298-f002:**
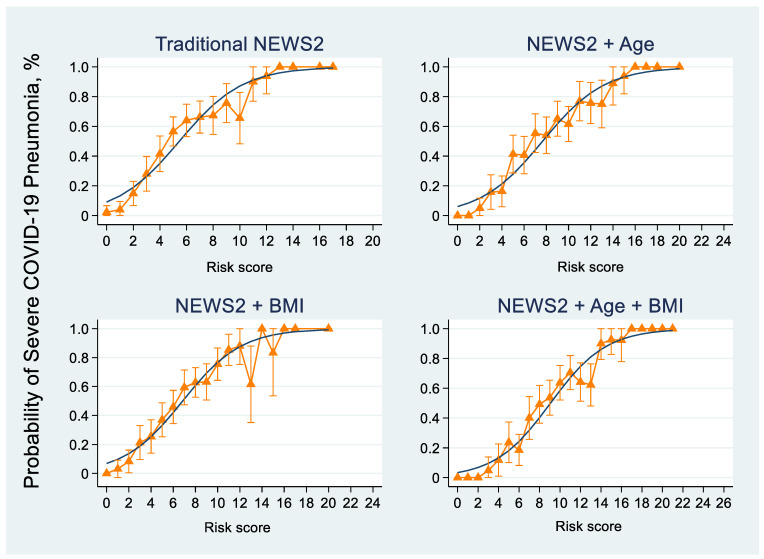
The predicted risk score (gray color line) and observed outcome (orange triangle with 95% CI) of traditional NEWS2, NEWS2 + Age, NEWS2 + BMI, and NEWS2 + Age + BMI models.

**Table 1 jcm-13-00298-t001:** The patients’ baseline characteristics.

Variables	All Cases (*n* = 725)	Severe (*n* = 350)	Non-Severe (*n* = 375)	*p*-Value
Baseline demographics				
Age (yr)	46.7 ± 17.4	53.0 ± 16.3	40.9 ± 16.5	<0.001
Male, *n* (%)	372 (51.3)	195 (55.8)	177 (47.2)	0.026
Body mass index (kg/m^2^)	26.6 ± 6.5	27.5 ± 6.6	25.8 ± 6.2	<0.001
Current smoker, *n* (%)	53 (7.3)	31 (8.8)	22 (5.8)	0.188
Pre-existing comorbidities, *n* (%)				
Hypertension	253 (34.9)	162 (46.4)	91 (24.3)	<0.001
Diabetes mellitus	89 (12.3)	58 (16.6)	31 (8.3)	0.001
Respiratory diseases	33 (4.5)	15 (4.3)	18 (4.8)	0.859
Chronic kidney disease	30 (4.1)	22 (6.3)	8 (2.1)	0.008
Heart diseases	27 (3.8)	16 (4.7)	11 (3.0)	0.264
Others	28 (3.8)	16 (4.6)	12 (3.2)	0.441
Onset before hospitalization (days)	3 (2, 5)	4 (2, 6)	3 (2, 5)	0.005
Vital signs				
Body temperature (°C)	37.1 ± 0.9	37.2 ± 1.0	37.0 ± 0.9	0.002
Heart rate (beats/min)	93 ± 18	91 ± 18	94 ± 18	0.040
Respiratory rate (breathes/min)	24 ± 6	26 ± 7	22 ± 5	<0.001
Systolic blood pressure (mmHg)	127 ± 20	129 ± 20	124 ±19	0.001
Diastolic blood pressure (mmHg)	78 ± 13	78 ± 13	78 ± 12	0.691
Pulse oximetry (%) at room air	93.4 ± 6.3	90.8 ± 7.8	95.3 ± 3.1	<0.001
NEWS2 *	5 (3, 7)	6 (4, 9)	3 (2, 5)	<0.001
Laboratory investigations				
Hemoglobin (g/dL)	13.3 ± 2.1	13.4 ± 2.0	13.3 ± 2.4	0.402
White blood cells (10^3^ cells/mm^3^)	7.3 ± 3.7	8.3 ± 4.1	6.4 ± 2.9	<0.001
Absolute lymphocyte count 10^6^ (/mm^3^)	1.26 ± 0.77	1.07 ± 0.66	1.48 ± 0.82	<0.001
Blood urea nitrogen (mg/dL)	13 (10, 19)	16 (12, 23)	11 (9, 15)	<0.001
Creatinine (mg/dL) *	0.8 (0.7, 1.1)	0.9 (0.7, 1.2)	0.8 (0.7, 1.0)	0.001
D-dimer (ng/mL) *	473 (322, 1019)	509 (367, 1058)	382 (262, 689)	0.001
C-reactive protein (mg/L) *	61.1 (17.4, 115.5)	75.8 (40.1, 134.9)	27.8 (7.7, 82.3)	<0.001
Procalcitonin (ng/mL) *	0.11 (0.06, 0.28)	0.13 (0.07, 0.3)	0.06 (0.04, 0.12)	<0.001
Lactate dehydrogenase (U/L) *	354 (260, 474)	413 (322, 559)	271 (219, 395)	<0.001
ESR (mm/hour) *	36 (19, 58)	40 (23, 58)	30 (14, 54.7)	<0.001
Multilobe involvement ^#^, *n* (%)	586 (80.8)	326 (93.1)	260 (69.3)	<0.001

Continuous data are presented as mean ± SD; otherwise, * denotes median (IQR). ^#^ Based on chest imaging. Abbreviation: ESR, erythrocyte sedimentation rate; NEWS2, National Early Warning Score 2.

**Table 2 jcm-13-00298-t002:** The patients’ treatments, outcomes, and complications.

Variables	All Cases (*n* = 725)	Severe (*n* = 350)	Non-Severe (*n* = 375)	*p*-Value
Treatments, *n* (%)				
Antiviral drugs				
Favipiravir	588 (81.2)	242 (69.3)	346 (92.3)	<0.001
Remdesivir	327 (45.2)	274 (78.5)	53 (14.1)	<0.001
Steroids	615 (84.8)	347 (99.1)	268 (71.5)	<0.001
Oral steroids	306 (42.2)	190 (54.3)	116 (30.9)	< 0.001
Systemic steroids	544 (75.1)	325 (92.9)	219 (58.6)	<0.001
Vasopressor	73 (10.1)	71 (20.2)	1 (0.3)	<0.001
Tocilizumab	68 (9.4)	68 (19.4)	0 (0.0)	<0.001
High-flow oxygen nasal cannula	306 (42.2)	306 (87.4)	0 (0.0)	<0.001
Mechanical ventilator	110 (15.2)	110 (31.4)	0 (0.0)	<0.001
Outcomes				
ICU admission	338/725 (46.6)	338/350 (96.6)	0/375 (0.0)	<0.001
ICU mortality	55/446 (12.3)	55/338 (16.3)	0/0 (0.0)	<0.001
Hospital mortality	57/725 (7.9)	57/350 (16.3)	0/375 (0.0)	<0.001
ICU length of stay	4 (7, 11)	8 (5, 12)	4 (2, 6)	<0.001
Hospital length of stay	9 (5, 13)	10 (6, 14)	8 (5, 13)	<0.001
Complication, *n* (%)				
ARDS requiring IMV	47 (6.5)	47 (13.4)	0 (0)	<0.001
Acute kidney injury	46 (6.3)	36 (10.3)	10 (2.7)	<0.001
HAP/VAP	57 (7.9)	39 (11.1)	18 (4.8)	0.002
Pulmonary embolism	8 (1.1)	6 (1.7)	2 (0.5)	0.164

Abbreviation: ARDS, acute respiratory distress syndrome; HAP/VAP, hospital-acquired pneumonia/ventilator-associated pneumonia; ICU, intensive care unit; IMV, invasive mechanical ventilation.

**Table 3 jcm-13-00298-t003:** The best multivariable prognostic factors for severe COVID-19 pneumonia.

Variables	Univariable OR (95%CI)	*p*-Value	Multivariable OR (95%CI)	*p*-Value
NEWS2	1.523 (1.424–1.629)	<0.001	1.459 (1.361–1.565)	<0.001
Age	1.044 (1.034–1.054)	<0.001	1.040 (1.028–1.052)	<0.001
Male	1.407 (1.050–1.886)	0.022	-	-
Body mass index	1.042 (1.018–1.067)	0.001	1.077 (1.044–1.110)	<0.001
Current smoker	1.196 (0.986–1.450)	0.069	-	-
Hypertension	1.603 (1.389–1.851)	<0.001	-	-
Diabetes mellitus	2.226 (1.408–3.521)	0.001	-	-
Respiratory diseases	0.888 (0.440–1.790)	0.740	-	-
Chronic kidney disease	3.084 (1.353–7.065)	0.007	-	-
Heart diseases	1.585 (0.725–3.464)	0.248	-	-
Other diseases	1.449 (0.676–3.108)	0.341	-	-

Abbreviation: NEWS2, National Early Warning Score 2.

**Table 4 jcm-13-00298-t004:** The best multivariable prognostic factors for severe COVID-19 pneumonia.

Variables	Univariable OR	*p*-Value	Multivariable OR	*p*-Value	Coefficient	Score
	(95%CI)		(95%CI)			
NEWS2	1.523 (1.424–1.629)	<0.001	1.459 (1.361–1.565)	<0.001	0.378	1
Age (yr)	1.044 (1.034–1.054)	<0.001	1.040 (1.028–1.052)	<0.001	-	-
<40	Ref					0
40–59	3.369 (2.349–4.832)	<0.001	2.827 (1.826–4.375)	<0.001	1.039	3
≥60	4.934 (3.325–7.324)	<0.001	4.211 (2.564–6.918)	<0.001	1.438	4
BMI (kg/m^2^)	1.042 (1.018–1.067)	0.001	1.077 (1.044–1.110)	<0.001	-	-
<24.9	Ref					0
25.0–29.9	1.560 (1.093–2.226)	0.014	1.909 (1.233–2.956)	0.004	0.647	2
≥30.0	1.799 (1.248–2.593)	0.002	2.819 (1.760–4.514)	<0.001	1.036	3

Abbreviation: BMI, body mass index; NEWS2, National Early Warning Score 2.

**Table 5 jcm-13-00298-t005:** The model performance of NEWS2 and NEWS2 Plus models.

Determinants	Apparent Model Performance				Bootstrap Internal Validation			
	NEWS2	NEWS2 + Age	NEWS2 + BMI	NEWS2 + Age + BMI	NEWS2	NEWS2 + Age	NEWS2 + BMI	NEWS2 + Age + BMI
Score range	0 to 20	0 to 24	0 to 23	0 to 27	0 to 20	0 to 24	0 to 23	0 to 27
C-statistic (95% CI)	0.798 (0.767–0.83)	0.811 (0.780–0.841)	0.800 (0.768–0.832)	0.821 (0.791–0.850)	0.797 (0.766–0.831)	0.810 (0.780–0.843)	0.800 (0.769–0.831)	0.821 (0.792–0.852)
Difference in C statistic (95% CI)	Ref	0.012 (−0.005–0.029)	0.001 (−0.013–0.016)	0.022 (0.005–0.040)	Ref	0.012 (−0.005–0.029)	0.001 (−0.013–0.016)	0.022 (0.005–0.040)
IDI (95% CI)	Ref	2.9 (0.9–5.7)	1.0 (0.0–2.8)	5.3 (2.9–8.7)	Ref	2.9 (0.5–5.3)	1.0 (−0.3–2.4)	5.3 (2.4–8.3)
NRI (95% CI)	Ref	44.4 (30.6–59.4)	22.1 (0.4–39.5)	45.7 (32.1–65.0)	Ref	44.3 (29.6–59.1)	22.1 (4.8–39.3)	45.7 (29.1–62.3)
Calibration intercept (95% CI)	0.105 (0.071–0.156)	0.068 (0.043–0.106)	0.074 (0.047–0.116)	0.035 (0.020–0.060)	0.106 (0.089–0.126)	0.068 (0.057–0.081)	0.075 (0.063–0.089)	0.035 (0.030–0.042)
CITL (95% CI)	0.000 (−0.172–0.172)	0.000 (−0.176–0.176)	0.000 (−0.172–0.172)	0.000 (−0.179–0.179)	−0.002 (−0.178–0.161)	−0.002 (−0.185–0.178)	−0.002 (−0.172–0.167)	−0.002 (−0.190–0.185)
CS (95% CI)	1.000 (0.837–1.163)	1.000 (0.843–1.157)	1.000 (0.838–1.162)	1.000 (0.845–1.155)	0.997 (0.838–1.168)	0.996 (0.860–1.152)	1.001 (0.857–1.176)	0.998 (0.865–1.154)

Abbreviation: BMI, body mass index; CS, calibration slope; CITL, calibration in the large; IDI, integrated discrimination improvement; NEWS2, National Early Warning Score 2; NRI, net reclassification improvement.

**Table 6 jcm-13-00298-t006:** The model performance of NEWS2 and NEWS2 Plus models.

Determinants	Score	Cut-Off	Sensitivity (%)	Specificity (%)	PPV (%)	NPV (%)
NEWS2	0 to 20	5	83.7 (79.4–87.4)	63.5 (58.4–68.5)	68.1 (63.5–72.5)	80.7 (75.5–85.0)
NEWS2 + Age	0 to 24	5	95.4 (72.7–97.4)	46.4 (41.3–51.6)	62.4 (58.2–66.5)	91.6 (86.7–95.1)
NEWS2 + BMI	0 to 23	5	91.7 (88.3–94.4)	48.0 (42.8–53.2)	62.2 (57.9–66.4)	86.1 (80.7–90.5)
NEWS2 + Age + BMI	0 to 27	5	99.7 (98.4–100)	26.7 (22.1–31.7)	57.7 (53.6–61.7)	98.9 (94.2–100)

Abbreviation: BMI, body mass index; NEWS2, National Early Warning Score 2; NPV, negative predictive value; PPV, positive predictive value.

## Data Availability

Data are unavailable due to privacy or ethical restrictions.

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
