# Peer review of "The National Early Warning Score 2 with Age and Body Mass Index (NEWS2 Plus) to Determine Patients with Severe COVID-19 Pneumonia"

_jcm, 2024, doi:10.3390/jcm13010298_

Round 1

Reviewer 1 Report

Comments and Suggestions for Authors

Abstract: line 25: In conclusion instead of Inconclusion.

Results: page 9 table 6: specificity of NEWS2 plus is less that of traditional NEWS2 system explain this?

If so, ways to improve specificity of the NEWS2 plus model.

Reviewer 2 Report

Comments and Suggestions for Authors

I read with great interest the manuscript by Trongtrakul et al. The study is sound and interesting. However, there are some issues that need to be addressed:

- Lines 33-39. Authors should report that COVID-19 disease does not only involve respiratory tract, but it may also impact cardiovascular function (doi: 10.1007/s00134-022-06685-2 - doi: 10.1007/s00134-023-07147-z), gastrointestinal (doi: 10.1093/trstmh/trab042) and cutaneous manifestations (doi: 10.1111/dth.13549), as well as coagulation disorders (doi: 10.1007/s12185-020-03029-y). Please briefly discuss and add these 5 references.

- Lines 94-97. Please use abbreviations only for word that are used more than three times throughout the paper.

- Line 120. Please remove this paragraph.

- In general the methods section is quite long and hard to read. Please remove the unnecessary information in order to increase readability.

- Please remove subparagraphs from the discussion section.

- Please include the fact that the registry is limited to one country as a limitation of the study.

Reviewer 3 Report

Comments and Suggestions for Authors

Dear author

1- Tables and examples are not clear for readers.

2- In terms of grammar, it should be checked again.

3- Does high weight affect the severity of corona infection? Why are scientific documents available?

4- What is the purpose of this study?

5- How can this scoring help in the treatment of Covid-19?

6- Can this rating and index be used for other respiratory diseases?

Comments on the Quality of English Language

* Extensive editing of English language required

Round 2

Reviewer 3 Report

Comments and Suggestions for Authors

Dear editor

The answer to the comments is not convincing.

Comments on the Quality of English Language

Extensive editing of English language required